# Computational Model Exploring Characteristic Pattern Regulation in Periventricular Vessels

**DOI:** 10.3390/life12122069

**Published:** 2022-12-09

**Authors:** Hisako Takigawa-Imamura, Saito Hirano, Chisato Watanabe, Chiaki Ohtaka-Maruyama, Masatsugu Ema, Ken-ichi Mizutani

**Affiliations:** 1Anatomy and Cell Biology, Graduate School of Medical Sciences, Kyushu University, 3-1-1 Maidashi, Higashi-ku, Fukuoka 812-8582, Japan; 2Yahata Kousei Hospital, 3-12-12 Satonaka, Yahatanishi-ku, Kitakyushu 807-0846, Japan; 3Department of Stem Cells and Human Disease Models, Research Center for Animal Life Science, Shiga University of Medical Science, Otsu 520-2192, Japan; 4Developmental Neuroscience Project, Tokyo Metropolitan Institute of Medical Science, Tokyo 156-8506, Japan; 5Laboratory of Stem Cell Biology, Graduate School of Pharmaceutical Sciences, Kobe Gakuin University, 1-1-3 Minatojima, Chuo-ku, Kobe 650-8586, Japan

**Keywords:** mathematical model, pattern formation, neocortical vasculature, angiogenesis, endothelial cell

## Abstract

The developing neocortical vasculature exhibits a distinctive pattern in each layer. In murine embryos, vessels in the cortical plate (CP) are vertically oriented, whereas those in the intermediate zone (IZ) and the subventricular zone (SVZ) form a honeycomb structure. The formation of tissue-specific vessels suggests that the behavior of endothelial cells is under a specific regulatory regime in each layer, although the mechanisms involved remain unknown. In the present study, we aimed to explore the conditions required to form these vessel patterns by conducting simulations using a computational model. We developed a novel model framework describing the collective migration of endothelial cells to represent the angiogenic process and performed a simulation using two-dimensional approximation. The attractive and repulsive guidance of tip cells was incorporated into the model based on the function and distribution of guidance molecules such as VEGF and Unc ligands. It is shown that an appropriate combination of guidance effects reproduces both the parallel straight pattern in the CP and meshwork patterns in the IZ/SVZ. Our model demonstrated how the guidance of the tip cell causes a variety of vessel patterns and predicted how tissue-specific vascular formation was regulated in the early development of neocortical vessels.

## 1. Introduction

Vascular endothelial cells (ECs) play fundamental roles in the outline of vessel shapes. The primary culture of ECs, derived from rodent aorta and human umbilical vein, has demonstrated that ECs have the ability to conduct angiogenic growth *in vitro* [1,2,3], indicating that the branching structure of vessels is intrinsically involved in ECs. ECs cooperatively migrate in elongated shapes and remain bound to neighboring cells [4]. Inspired by these findings, we hypothesized that branched pattern formation might be explained by the collective migration of elongated cells that are aligned together or displaced. Meanwhile, the behavior of ECs is regulated by various signaling guidance, and it is important to investigate how guidance events control angiogenic patterns, so as to understand the initial step for tissue-specific vessel construction. In this study, we examined how tissue-specific vascular patterns are formed by focusing on angiogenesis during early brain development.

The brain is an interesting tissue where diverse vascular patterns can be observed. In particular, the developing neocortex is a precisely organized tissue with a layered structure generated by specific cellular arrangements and molecular expression [5]. Each layer is composed of different neural cell populations, which has implications for brain formation and function, and exhibits a characteristic pattern of vascular cell orientation: vertically oriented vessels in the cortical plate (CP) without branching, and a honeycomb pattern in the subventricular zone (SVZ) and intermediate zone (IZ) with higher branching during the developing neocortex (Figure 1a) [6]. Although the different functional characteristics of ECs that distinguish region-specific properties remain poorly understood, a specific vascular environment is utilized to regulate brain development and formation.

The synchronized development of the vascular and nervous systems is guided by common molecules that communicate with each other in neurovascular crosstalk. In this context, guidance ligands and their receptors have been shown to regulate the differentiation of both systems. For example, in the development of the neocortex, guidance molecules such as netrin and semaphorin are known to mediate neuronal migration and axonal elongation, which have also been shown to play similar roles in angiogenesis [7,8]. In addition, neural cells localized to the SVZ and IZ express the netrin receptor Uncoordiated-5d (Unc5d) and fibronectin leucine-rich transmembrane protein (FLRT) family, which exhibit repulsive activity in cell migration [9]. In terms of EC-specific attractive guidance, vascular endothelial growth factor (VEGF) is a key angiogenic factor with promitogenic and promigratory effects [10]. VEGF is predominantly expressed on the ventricular surface [6], and the molecular interactions that characterize each region may also correspond to differences in vascular patterns; however, it is not clear how the orchestration of these signals contributes to the characterization of the functional morphology of vessels.

Various mathematical models have shown how ECs form branches in response to angiogenic factor distribution such as VEGF [11,12,13,14]. Angiogenic induction in tumors by chemical stimuli was described by a biased random walk of ECs to explain their dense branched structure [11]. The contribution of cell motility and proliferation to branch formation has been examined using a phase-field framework [13]. Regarding meshwork formation, the vasculogenic process seen in the *in vitro* culture of human umbilical vein endothelial cells (HUVECs) was explained using a cellular Potts model [12]. However, the manner in which organ-specific vascular patterns are formed has not been well investigated in mathematical modeling studies. The effect of repulsive guidance on vascular patterns remains underexplored.

In this study, we developed a computational model describing the migration and interaction of ECs to explore the conditions for generating the characteristic vascular patterns of the CP and IZ/SVZ vessels. First, our model expresses cell-autonomous angiogenic growth of branching vessels. Second, we showed how the external guidance of ECs controls a variety of vascular morphologies. Finally, we discuss how each attribute of the modeled ECs affects their patterning.

## 2. Materials and Methods

### 2.1. Mice and In Vivo Experiment

*In vivo* experiments were performed in strict accordance with the recommendations in the Guide for the Care and Use of Laboratory Animals of Kobe Gakuin University and Tokyo Metropolitan Institute of Medical Science. The protocol was approved by the Committee on the Ethics of Animal Experiments of Kobe Gakuin University and Tokyo Metropolitan Institute of Medical Science. ICR mice were obtained from Japan SLC (Kobe, Japan). *Flt1* (*VEGFR1*)-tandem dsRed bacterial artificial chromosome transgenic mice have been described previously [15].

*In utero* electroporation was performed as previously reported [16]. Timed pregnant wild-type ICR mice were anesthetized, and 0.5 μL of a mixture of plasmid DNA, including pCAG-Kir2.1 target plasmid and pCAGGS-EGFP reporter plasmid, was directly injected into the lateral ventricles of the embryonic forebrain using a glass micropipette. *In utero* electroporation at embryonic day 10.5 (E10.5) was performed using an electroporator (CUY21E, BEX), the embryonic brains were dissected at E15.5, and fixed in 4% paraformaldehyde (PFA)/PBS overnight at 4 °C.

### 2.2. Immunohistochemistry and Imaging Analysis

To observe the three-dimensional patterns of developing blood vessels, 150 μm thick coronal sections were prepared from both *VEGFR1*-tdsRed telencephalon. Briefly, brains were fixed in 4% PFA for 1 h, embedded in agarose gel, and cut using a vibrating microtome (Leica VT1200). Vibratome sections were treated with blocking buffer (10% donkey serum and 0.3% Triton X-100; pH 7.4) overnight at 4 °C, followed by incubation with primary antibodies diluted in the same buffer overnight at 4 °C. Immunolabeled sections were washed three times in 0.3% Triton X-100 (PBST) for 30 min and incubated with secondary antibodies overnight at 4 °C. After washing, sections were mounted under a coverslip with mounting medium (Fluoro-KEEPER Antifade Reagent, Nacalai). Images were acquired on a confocal microscope (FV3000, Olympus). Images were processed using Adobe Photoshop software.

### 2.3. Numerical Simulations

The computer program was written in C++ and the calculations were performed using the Linux system. Calculation results were visualized using Mathematica software (Wolfram Research).

The inter-vessel spacing in the CP vessel pattern results was calculated by dividing the areas of gaps between branches by the pattern height. The inter-vessel spacing in the IZ/SVZ vessel pattern was obtained as the minor axis length of the fitted ellipse of the mesh hole using Fiji software [17]. Statistically significant differences in the spacing between the patterns were evaluated by two-tailed Welch’s *t*-test using R software.

## 3. Modeling

### 3.1. Concept and Rational of EC Model

The structural difference between the vertical extension of CP vessels and the honeycomb network of IZ/SVZ vessels in the developing neocortex can be compared to the difference between parallel straight branches and the meshwork in two-dimensional space, where the *x*-axis represents the ventral-dorsal axis and the *y*-axis represents the ventricular-cortical axis (Figure 1b). We considered a two-dimensional system in which ECs autonomously migrate and bind to each other to form vessels as a representation of angiogenic growth. Agent-based modeling was used to describe the collective migration of ECs. The intrinsic cell-autonomous behaviors were assumed as follows:ECs have an elongated shape and migrate along their longitudinal axis (Figure 2a);Cell–cell binding affects cell orientation and migration (Figure 2b);ECs divide randomly.

The model consisted of two types of ECs: *tip cells* and *stalk cells*. The cell extending many motile filopodia at the leading edge of the vascular branch is termed the tip cell and is distinguished from subsequent stalk cells [18,19]. Tip cells are thought to respond to external guidance cues in order to undergo directional and coordinated vessel growth [18,20,21]. Tip cells were assumed to be distinguished from stalk cells by their relative positions; cells at the forefront of the vessels were tip cells (Figure 2a). It was assumed that directional noise on EC migration could cause sprouting from a preexisting vessel and splitting of a growing forefront [4,22,23] and generate a new tip cell. We omitted the process of tip cell selection, which is mediated by VEGF and Delta-like 4 (DLL4)-Notch signaling [24,25]. This assumption can be summarized as follows:4.Branch formation occurs by chance.

Tissue-specific vascular patterns are regulated by the external guidance of tip cells [18,26]. We incorporated attractive and repulsive guidance into our model. VEGF is specifically expressed in the VZ of the developing neocortex [6]. We considered the possibility that a concentration gradient of VEGF was formed throughout the neocortex, which exerts chemotactic attraction toward the direction of the VZ. Regarding repulsive guidance, the Unc receptor mediates axonal guidance via its ligands such as netrins and FLRTs, and ECs are also thought to be under this regulation [9,27]. For instance, Netrin-1 has been shown to have a negative role in tip cell guidance by filopodial retraction, and knockout of its receptor Unc5b resulted in excess branching of brain capillaries in embryonic mice [28]. It is not yet clear how ligands express, are distributed, and provide guidance to tip cells; thus we adopted the most primitive assumption within the scope of the model framework that tip cells receive a repulsive force from other ECs. We now have two additional assumptions regarding the extrinsic regulation:5.Tip cells may undergo directed vertical migration by chemotaxis;6.Tip cells may receive repulsive guidance from other ECs.

Repulsion was defined as acting only over long distances and not between close neighboring cells (Figure 2c), so that repulsion does not impede anastomosis, which is one of the essential events in angiogenesis [29,30,31]. The anastomosed tip cell switched to a stalk cell in the model (Figure 2a).

The contributions of other signaling interactions, local blood flow, and cell types other than ECs such as astrocytes [26] and pericytes [32] were omitted from the model.

### 3.2. Construction of the EC Model

Based on the above assumptions, we constructed a model tentatively named the EC model. All cell shapes in the model were simplified rectangles of length l and width w (Figure 2a,d, Assumption 1). The midpoint of the front edge was defined as *the head*, that of the rear end as *the tail*, and their positions as $h=\left(h_{x},h_{y}\right)$ and $r=\left(r_{x},r_{y}\right)$, respectively (Figure 2d). It was assumed that each cell moves from the head at a constant velocity *v* = 0.4, as long as its contact position with the subsequent cell is $<l_{c}=1.4$ from the head, to prevent the cell from being detached from the cell group (Assumption 2).

The tip and stalk cells were assumed to have different modes of migratory regulation. Tip cells lead to branch elongation, whereas stalk cells move along the cells in contact.

The migratory direction of the stalk cell $V^{\mathrm{stk}}$ was calculated based on the cell orientation u=h−r/l as follows:(1)$$V^{\mathrm{stk}}=\frac{u+k_{a}a+\gamma \eta }{\left|u+k_{a}a+\gamma \eta \right|}\;,$$
where $k_{a}$ and γ are the magnitudes of the cell alignment with neighboring cells and small directional noise, respectively. The alignment direction $a_{i}$ was introduced such that the cell was progressively aligned to its neighbors (Figure 2b, Assumption 2) as follows:(2)$$a_{i}=\frac{\sum_{j\in Z_{i}}(H_{j}-r_{i})}{\left|\sum_{j\in Z_{i}}(H_{j}-r_{i})\right|}\;,$$
where $Z_{i}$ is a set of cells that precede and contact cell *i*, and $H_{j}$ is either $h_{j}$ or $r_{j}$, which is further forward from cell *i*. Cell–cell contact was determined by whether any side of the rectangular cell had an intersection with any side of the other cell. $Z_{i}$ is also used as a criterion for tip cells as cell *i* is the tip cell if $n\left(Z_{i}\right)=0$ (Figure 2a).

The noise $\eta_{i}$ was defined as a random rotation of $u_{i}$ as follows:(3)$$\eta_{i}=u_{i}\left(\begin{array}{cc} \cos\theta & -\sin\theta \\ \sin\theta & \cos\theta \end{array}\right)\;,$$
where θ is a uniform random number in the interval [−π/4,π/4] in this study. When a stalk cell is sufficiently displaced from the cell branch by noise, it turns into a tip cell, which may follow the initial step of sprouting from a preexisting vessel (Assumption 4).

The migratory direction of the tip cells $V^{\mathrm{tip}}$ is determined by the intrinsic noise and external guidance as follows:(4)$$V^{\mathrm{tip}}=\frac{u+k_{c}c+k_{b}b+\sigma \eta }{\left|u+k_{c}c+k_{b}b+\sigma \eta \right|}\;,$$
where $k_{c}$, $k_{b}$, and σ represent the directed migration, repulsion, and directional noise for tip cells, respectively (Assumption 4). The forced direction was basically set $c=\left(0,-1\right)$ for vertical chemotaxis (Assumption 5). Long-range repulsion between ECs (Assumption 6) was introduced so that the cell moved in a direction that repels neighboring cells as follows:(5)$$b_{i}=\left(\underset{j\in Q_{i}}{\sum }\frac{h_{j}-h_{i}}{\left|h_{j}-h_{i}\right|}\right)/\left|\underset{j\in Q_{i}}{\sum }\frac{h_{j}-h_{i}}{\left|h_{j}-h_{i}\right|}\right|\;,\;$$
where $Q_{i}$ is a set of cells whose head is contained in a region between the inner diameter *Ri* and the outer diameter *Ro* centered on $h_{i}$. Directed migration and cell–cell repulsion acting on tip cells represent the external guidance for EC migration, in contrast to cell alignment and noise being cell-autonomous behaviors.

We introduced noise for tip cells σ to describe the random process in which tip cells sporadically split a growing forefront. The value σ basically takes the same value as the basic low noise γ, though this switches to a constant large value $\bar{\sigma }\;$with probability Pσ. In addition, the cell orientation is reversed with the probability of *Pf* by switching the head and tail [4], which enables the model to represent smooth anastomosis and tip cell exchange [25].

In the update of the cell position, we assumed that the tail moves forward along **u** and the head moves such that the cell orientation becomes **V,** as follows:(6)$$\begin{array}{c} r_{i}\left(t+\Delta t\right)=r_{i}\left(t\right)+v\Delta tu_{i}\\ h_{i}\left(t+\Delta t\right)=r_{i}\left(t+\Delta t\right)+lV_{i}\;. \end{array}$$

We considered the progress of angiogenesis in rectangular areas with a height of 20 and a width of 47 (CP) or 55 (IZ/SVZ). If the head exits the area, the migratory direction (6) is modified as follows:(7)$$h_{i}\left(t+\Delta t\right)=r_{i}\left(t+\Delta t\right)+l(V_{i}+q)/\left|V_{i}+q\right|\;,$$
where **q** is the inward normal vector to the area boundary.

Tip and stalk cells were assumed to proliferate with probabilities *Pt* and *Ps*, respectively (Assumption 3). Cell proliferation was described by adding new cells. The position of the new cell *x* was assigned based on the position of the original cell *i* as follows:(8)$$\begin{array}{c} h_{x}=h_{i}+du_{i}\\ r_{x}=r_{i}+du_{i}\;, \end{array}$$
where *d* is the distance of the divided cells.

We set l=2, w=0.4, γ=0.01 $\bar{\sigma }=60$, Pσ=0.01, Δt=0.01, d=0.3, and other parameter values are shown in Figure legends.

## 4. Results

### 4.1. CP Vessel Pattern

Numerical simulations were performed to determine whether the EC model could explain vascular pattern formation. First, the growth of a cell population via cell-autonomous behavior was examined. One cell was set as the initial state and was fixed at the position (Figure 3a). The cells randomly proliferated and migrated, then the cell group formed a curved branching structure (Figure 3b). The cell branch is inclined to extend along the direction of the initial cell owing to the effect of the cell alignment. As the number of cells increased, the orientation of the initial cell lost its influence. Splitting of a growing forefront can occur because of high noise that is sporadically generated in a tip cell. To suppress the effect of noise, we introduced the external guidance of vertical directed migration in (4). The branch then elongated linearly, which is reminiscent of the linear pattern of CP vessels (Figure 3c).

Subsequently, we attempted to reproduce a group of CP vessels. CP vessels have been reported to grow from the plexus of pial vessels [6,33]. As an initial state to represent the pial vessel plexus, we set a horizontal sequence of randomly oriented cells fixed at these positions (Figure 3d). When ECs were grown in the model, directed migration induced parallel branch extension but with dense spacing (Figure 3e). Next, we added another external guidance, cell–cell repulsion, and the space between parallel branches increased (Figure 3f). The inter-vessel spacing was 3.54±0.087 for the repulsive range *Ri* = 4 and *Ro* = 5. The inter-vessel spacing in the model was appropriate compared to the model cell size l=2, as the vessel spacing of CP vessels *in vivo* was in the order of 15–20 µm and the EC size ~10 µm [6]. The vessels were bent horizontally near the border of the angiogenic area and formed a plexus-like structure reminiscent of the subplate (SP) vessel pattern. We confirmed that the exclusion of external guidance resulted in the generation of winding branches with branching and looping (Figure 3g), indicating that directed migration and repulsion were essential for the realization of the CP vascular pattern. This result also demonstrated the potential of the model to build a variety of architectures with cell-autonomous behavior.

### 4.2. IZ/SVZ Vessel Patterning

Vessels in the IZ and SVZ are developed by vasculature extension from the CP vessels through the SP towards the ventricular side and from the ventral side to the dorsal side as neocortical development progresses [6,33]. To reproduce the formation process of the IZ/SVZ vessel pattern, 11 initial cells were arranged downward on the upper border and one cell was positioned inward on the left border (Figure 4a). The combination of cell-autonomous behaviors and repulsive effects successfully generated a meshwork pattern (Figure 4b). The cell branches bifurcated by chance and folded back at the edge, resulting in the formation of a web covering the entire region. The inter-vessel spacing was 3.9±0.44 for the repulsive range (*Ri*, *Ro*) = (4, 5). The initial pattern did not affect the formation of the meshwork or the inter-vessel spacing (4.0±0.89, Figure 4c,d and Appendix A), while the absence of repulsion had a significant impact on the pattern (Figure 4e). The cell branches formed fewer bifurcations and did not cover the region without repulsion. We further examined how repulsion contributes to the pattern by changing the repulsive range, and confirmed that the meshwork size depended on *Ri* and *Ro* (Figure 4f–i and Appendix A). The case (*Ri*, *Ro*) = (2, 5) shows a significantly smaller pattern compared to the case (*Ri*, *Ro*) = (4, 5), whereas no significant difference was found between the cases (*Ri*, *Ro*) = (4, 7) and (*Ri*, *Ro*) = (4, 5) (Appendix A), suggesting that the shortest distance at which repulsion is exerted is critical for the pattern size. This model was also shown to be able to generate patterns of inter-vessel spacing comparable to that of IZ/SVZ vessels on the order of 20–40 µm in vivo [6].

### 4.3. Effect of Impaired SP Function

As we have shown so far, our model suggests that distinctive regulation regimes of angiogenesis progress in the CP and IZ/SVZ bounded by SP. Migrating neurons receive specific differentiation controls for the multipolar-to-bipolar transition when passing through the SP [34]. When the excitatory activity of SP neurons was repressed by overexpressing the potassium channel Kir2.1, radial migration from the IZ to the CP was blocked [16]. Here, we observed how impaired SP function and neuronal migration affected vascular patterning. Kir2.1 overexpression resulted in increased vascular branching and frequent occurrence of abnormal EC clusters in both the VZ/SVZ/IZ and CP instead of the characteristic plexus (Figure 5). Quantification of the vascular branch points for the VZ/SVZ/IZ and SP/CP regions showed that genetic manipulation significantly increased the vascular branching per area in both regions (Figure 5g), suggesting that the precise regulation of neural development is an important factor determining the quality and quantity of tissue vascularization.

We considered what kind of changes in EC behavior explain the effect of SP neurons deficiency. We focused on structural changes in the CP, which are also linked to the difference between the CP and IZ/SVZ patterns, and explored which parameters in the model affected branch formation. In numerical simulations, the levels of directed migration $k_{c}$ and repulsive interaction $k_{b}$ differed greatly between the CP model and the IZ/SVZ model. Accordingly, we examined how these parameters control vascular patterning by simulating a simple angiogenic process initiated from a single EC under horizontally directed migration (Figure 6). If the level of directed migration is stronger than that of repulsion, a single vessel extends linearly without branching. The balance between directed migration and repulsion generates equally spaced horizontal vessels. As repulsion dominates, the spreading of the vessels becomes isotropic and the number of branches increases. Based on this result, we examined CP vessel simulation by changing the values of $k_{c}$ and $k_{b}$, however, no clear changes from linear patterns to branched meshwork-like patterns were observed (Appendix A).

We further selected five parameters that are likely to be involved in branch formation (Appendix A). By simulating the angiogenic process while varying each value, we found that the direction of vessel growth varied with increased branching when cell alignment $k_{a}$ was small and when the tip noise frequency Pσ was large. Cell alignment and tip noise frequency positively and negatively mediate coordinated cell migration, respectively. Therefore, we examined CP vessel simulation by changing the values of $k_{c}$ and $k_{b}$ under small $k_{a}$ and large Pσ conditions (Figure 7). It was found that a meshwork was formed when $k_{c}=0.03$ and $k_{b}$ was adjusted. It should be noted that low-density and high-density patterns were obtained in the cases $k_{b}=0.1\;$and $k_{b}=0.4$, respectively, implying that the level of repulsive guidance is another candidate for controlling vessel density. In contrast, CP vessel patterns were observed for higher $k_{c}$ and $k_{b}$. Taken together, pattern transition from the CP vessel to the branched meshwork requires organized regression of directed migration, inter-vessel repulsion, and coordinated cell migration.

## 5. Discussion

In this study, we decomposed the emergent complex vascular morphology process into intrinsic collective migration and tissue-specific extrinsic guidance. To the best of our knowledge, this is the first theoretical study to demonstrate a variety of angiogenic processes, whose orientation and density are regulated by the attractive and repulsive guidance of tip cell migration. Much remains to be elucidated regarding the molecular basis of vascular guidance for understanding neocortical angiogenesis, including diffusion signal dynamics, cellular responsiveness, EC-EC interactions, and neurovascular cell communication [8,24]. In the current situation, it may be helpful to demonstrate the possible association between EC guidance and vascular morphogenesis, leading to working hypotheses. Our agent-based model separately describes the behaviors of tip and stalk cells, and thus is applicable for incorporating notions from future findings.

Our model did not include the entire process of periventricular vessel formation. We assumed a static growth area and described angiogenic invasion, despite blood vessels invading as the tissue expands during neocortical development [6]. Vascular development cannot be isolated from the cooperative growth of surrounding tissues, which may dynamically generate physical ruts and chemical distributions for cell migration. *In silico* experiments are well suited for extracting the behavior of ECs to highlight qualitative differences in vasculogenic control in the CP and IZ/SVZ. The present results showed that CP vessel formation requires strong directional control and that meshwork formation requires attenuation of the coordinated cell migration and tuning of repulsion, which is valid for situations in which the tissue is expanding (Figure 8).

We introduced a repulsive interaction between ECs based on the finding that tip cells respond to a diffusible repulsive ligand common to axonal growth cones [2]. Reports that soluble repulsive guidance cues such as Netrin-1, Netrin-4, and Sema3 are expressed in ECs support the model assumptions [35,36,37,38], whereas little is known concerning their dynamics during neocortical development. It has also been shown that repulsive cues serve as attractive cues, depending on the cell types and receptors involved [7]. Context-dependent repulsive interaction may be related to the assumption in this study that repulsive forces do not work at close range, but only over longer distances. Alternatively, the repulsion assumption can be interpreted as a description of tip cells migrating to areas with fewer ECs. Stalk cells express the soluble receptor sVEGFR1, which traps VEGF, causing VEGF depletion in the vicinity of stalk cells [2]. Since tip cells express the membrane protein VEGFR2 and not sVEGF1, repulsive effects may not be exerted between tip cells. This may also be involved in the rationale behind the lack of repulsive effects at close range.

We proposed two candidates that may explain the difference in meshwork size in the IZ and SVZ. One is the distance at which the repulsive force begins to operate (Figure 4 and Appendix A) and the other is the strength of the repulsive force (Figure 7). It has been shown that FLRT2 is predominantly expressed in the CP, and its receptor Unc5d in the SVZ at E15.5 in mouse [9]. The SVZ and CP are tissues in which repulsive interaction was assumed in the model, while the low density of vessels seen in the IZ cannot be explained by the spatial distribution of these repulsive factors. In the CP and SVZ, the expression patterns of ligands and receptors are different, leaving the question of how repulsive interactions are mediated.

This study focused on the distinctive angiogenic processes in the developing neocortex. The structural difference between the vertically oriented vessels in the CP with lower branching and the honeycomb-patterned vessels in the IZ/SVZ with higher branching is strictly constructed by the boundary of SP neurons. We showed that the impaired SP neuronal activity influenced precise regulation of angiogenesis patterning (Figure 5). Although the mechanism by which inhibition of SP neuronal activity alters vascular patterns is not yet elucidated, migrating neurons that receive signals from SP neurons may affect vascular cell migration by altering signals in characteristic extracellular matrix such as proteoglycan localized in the SP layer. These results suggest a close interaction between neurogenesis and angiogenesis in the developing neocortex. Further investigations regarding how repulsive factors in neural cells control vascular cell behavior in each tissue will lead to an understanding of the regulatory mechanisms of development of neocortical vessels.

The model demonstrated that repulsion between ECs enlarged the branching angles, resulting in efficient space-filling (Figure 4b,d). This effect is caused by the repulsion to enlarge the difference in the directions of a preexisting vessel and a tip cell after intense tip noise occurs. The repulsive force generates the torque on tip cells to enlarge the branching angle after the branched structure appeared, whereas the initiation of branch formation itself is due to the stochastic rotation of tip cells. The angiogenic network could not spread over the area without repulsive interaction as shown in Figure 4e, where noisy migration often broadened branches and failed to form additional branches. This study suggests the existence of a mechanism to regulate branching angles when angiogenic sprouting occurs. The bifurcation angle of angiogenic sprouting affects tissue coverage efficiency, as the angle π/6 is theoretically the most efficient for covering an area per vascular length [39]. Our numerical simulations indicate that repulsive interaction involving ECs is important not only for pattern-scale regulation, but also for the construction of the network structure (Figure 3g and Figure 4e).

VEGF has been observed to be expressed locally in the VZ at E13.5 and shifted to the IZ at E15.5 in the developing mouse neocortex [6]. This notion indicates that the direction of the VEGF gradient is fluctuating during these stages, supporting the model assumption that the directed migration by chemotaxis was not involved in the IZ/SVZ model. In the CP model, ECs were reasonably assumed to be consistently chemotactic toward the ventricular side.

Cell alignment and tip noise frequency changed the direction of vessel branches, whereas tip noise intensity and cell division rates increased branching only (Appendix A). Even if a large angle change occurs in a tip cell owing to the high tip noise intensity, the branch angle gradually returns to horizontal owing to chemotaxis. On the other hand, if the frequency is high, new tip noise occurs and enlarges the branch angle before it returns to the horizontal. A high cell division rate increases the chance that stalk cells will accidentally deviate from preexisting vessels to become tip cells. This tendency was demonstrated in a previous study using a phase-field framework in which vascular thickness is necessary for the emergence of tip cells [13]. It has been suggested that stalk cells tend to proliferate more than tip cells [18], though we used the condition *Pt* = *Ps* in the CP vessel patterns and *Pt* > *Ps* in the IZ/SVZ patterns. Tip cell proliferation in this model also involves rapid migration of the tip cell because a new cell is inserted in front of the tip cell. As a future improvement to this model, it will be necessary to consider the assumption that proliferation occurs only in stalk cells and that tip cells are highly motile. VEGF is known to promote EC proliferation and induce tip cell selection through its involvement in Notch signaling [40]. Future modifications of the model will be examined to determine how tip cell selection affects angiogenic patterns.

## 6. Conclusions

We propose a novel computational model of angiogenic growth by describing the migration and proliferation of endothelial cells. The present study revealed that the positive and negative guidance of tip cells has the potential to construct a variety of vessel patterns. This study provides a working hypothesis for the contribution of VEGF gradients and repulsive guidance in the early development of neocortical vessels.

## Figures and Tables

**Figure 1 life-12-02069-f001:**
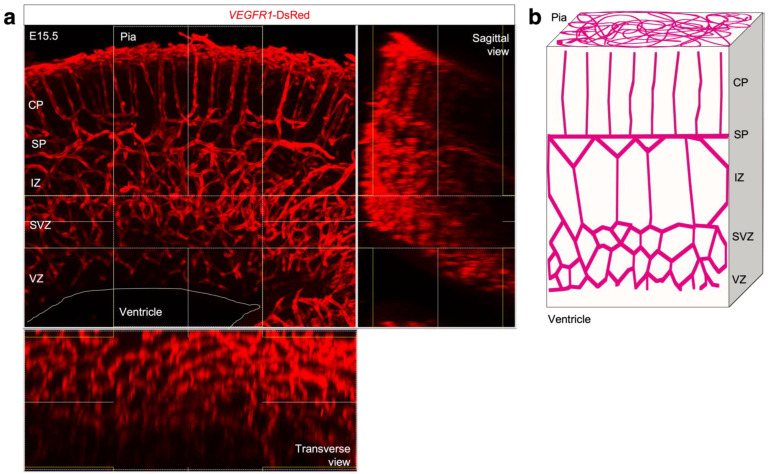
Periventricular vessels in developing neocortex. (**a**) Confocal z stack image of 150-µm section from VEGFR1-tdsRed neocortex at E15.5. Right and lower panels depict orthogonal views of the 3D reconstruction. (**b**) Schematic representation of the vascular pattern.

**Figure 2 life-12-02069-f002:**
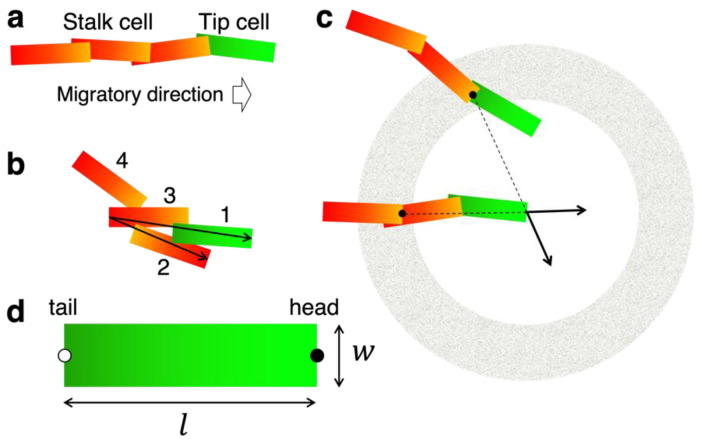
The essence of our computational model. (**a**) Vascular endothelial cells in the two-dimensional model. The tip cell (green) is at the top and stalk cells (red) are in the shank of the cell branch. For each cell, a traveling top (*head*, light green or orange) and a trailing end (*tail*, dark green or red) were defined. The overlap of cells is allowed in the model. (**b**) Cell alignment. In this example, cell 3 is aligned to cells 1 and 2, and is guided in the direction that is the summation of vectors indicated by arrows. Cell 4 is excluded from the align partners, which is located behind cell 3. (**c**) Repulsive guidance. Gray zone is the repulsive region. Cells whose heads are in this region (two cells with black dots in this example) exert a repulsive force on the tip cell in the center. The summation of vectors indicated by arrows is the repulsive guidance for the tip cell. (**d**) Definition of cell shape. The shape of a cell is a rectangle of length l=2 and width w=0.4 in this study. The midpoint of the front edge is head (closed circle), and that of the rear end is tail (open circle).

**Figure 3 life-12-02069-f003:**
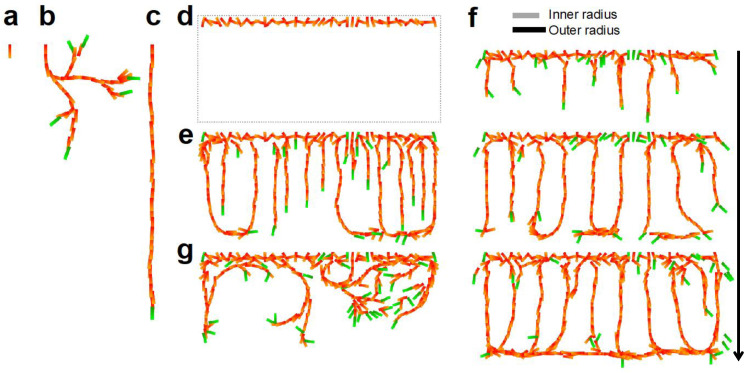
Numerical simulations of vessel patterns in our EC model under two-dimensional approximation. (**a**–**c**) Cell branch grown by cell-autonomous behaviors. The initial cell was placed downward as shown in (**a**). The branch extending from the cell bifurcated due to the sporadic large noise under the default condition $k_{c}=0$ in (**b**). Vertically forced migration $k_{c}=3$ made the branch straight in (**c**). *t* = 170. (**d**–**f**) The cortical plate vessel patterns. Initial cell arrangement and the area border was set as in (**d**). The blue area in (**d**) is a region where cells do not undergo directed migration. Vertically directed migration with $k_{c}=3$ generated branches extending vertically in dense at *t* = 100 in (**e**). When directed migration $k_{c}=3$ and repulsion $k_{b}=3$ were assumed, spaced sprouting reminiscent of the CP vessel pattern was reproduced in (**f**). Snapshots of *t* = 50, 100, and 150 are shown. The repulsive range was set as *Ri* = 4 and *Ro* = 5 as indicated by gray and black lines. (**g**) No external guidance $k_{c}=0\;$and $k_{b}=0$ resulted in a random branching pattern at *t* = 100. $k_{a}=0.004$, *Pf* = 0, *Pt* = 0.0002, and *Ps* = 0.0002.

**Figure 4 life-12-02069-f004:**
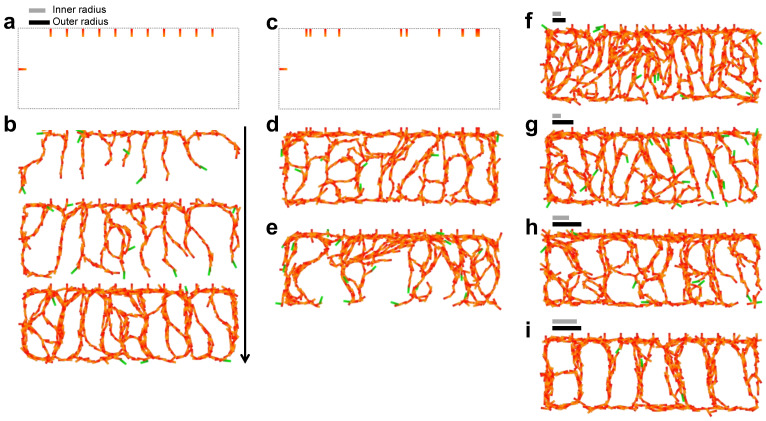
Meshwork patterns as representations of IZ and SVZ vessels. (**a**,**b**) Angiogenic process explained by the EC model. Dotted line indicates the area border, and the upper initial cells are arranged at equal intervals in (**a**). The repulsive range was set to *Ri* = 4 and *Ro* = 5 as indicated by gray and black lines. Snapshots of *t* = 0 (upper panel), *t* = 80 (second panel), *t* = 160 (third panel), and *t* = 240 (bottom panel) are shown in (**b**). No forced directional migration was assumed as $k_{c}=0$. $k_{a}=0.0006$, $k_{b}=0.01$, *Pf* = 0.05, *Pt* = 0.0016, and *Ps* = 0.0002. (**c**,**d**) Random initial arrangement (**c**) and the emergent pattern at *t* = 240 (**d**). Parameters were as in (**b**). (**e**) No repulsion as $k_{b}=0$. Other parameters were as in (b). (**f**–**i**) Meshwork sizes depending on the repulsive range. *Ri* = 2, *Ro* = 3 (**f**), *Ri* = 2, *Ro* = 5 (**g**), *Ri* = 4, *Ro* = 7 (**h**), and *Ri* = 6, *Ro* = 7 (**i**), as indicated by gray and black lines. Other parameters were as in (**b**).

**Figure 5 life-12-02069-f005:**
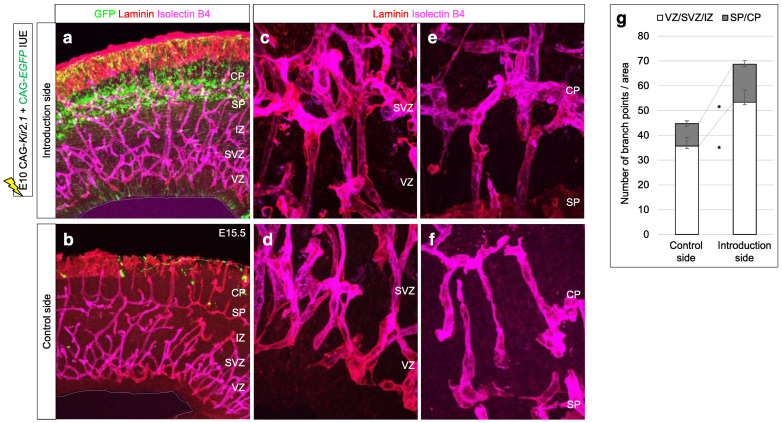
Experimental results of suppressing normal SP formation. (**a**–**f**) *In utero* electroporation of Kir2.1 gain-of-function vector with pCAG-EGFP was performed at E10.5 and analyzed at E15.5. Distribution of EGFP-positive cells was confirmed in the introduction side (**a**), although the opposite side of the neocortex (control side) had almost no EGFP signal (**b**). High-power images of immunofluorescence with Laminin and Isolectin B4 showing the vertically oriented vessels in the CP (**e**,**f**) and the honeycomb-patterned vessels in the SVZ/IZ (**c**,**d**). (**g**) The number of branch points per area was quantified in the control and introduction side; * *p* < 0.05, *n* = 5 independent experiments.

**Figure 6 life-12-02069-f006:**
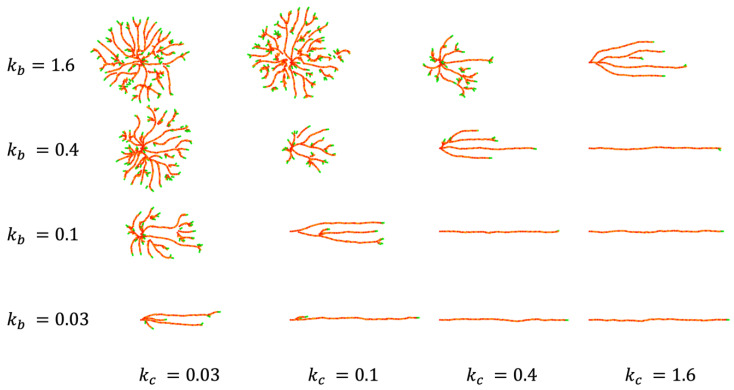
Branch formation and linear extension controlled by directed migration $k_{c}$ and inter-vessel repulsion $k_{b}$. The initial cell was placed with the migratory direction to the right. The forced direction was set $c=\left(1,0\right)$. Calculation results at *t* = 240 are shown. Other parameter values are as in Figure 3.

**Figure 7 life-12-02069-f007:**
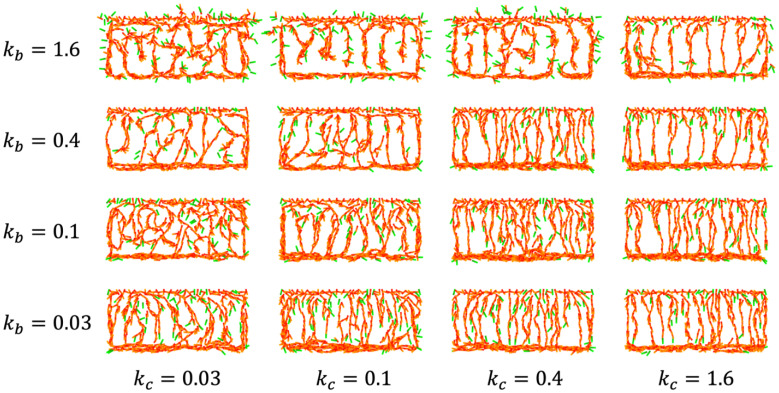
Parameter dependency of patterns in CP vessel simulation. The conditions were as in Figure 3f except for $k_{a}=0.002$, Pσ=0.03, *k_c_* and *k_b_* (indicated in the figure). Calculation results at *t* = 170 are shown.

**Figure 8 life-12-02069-f008:**
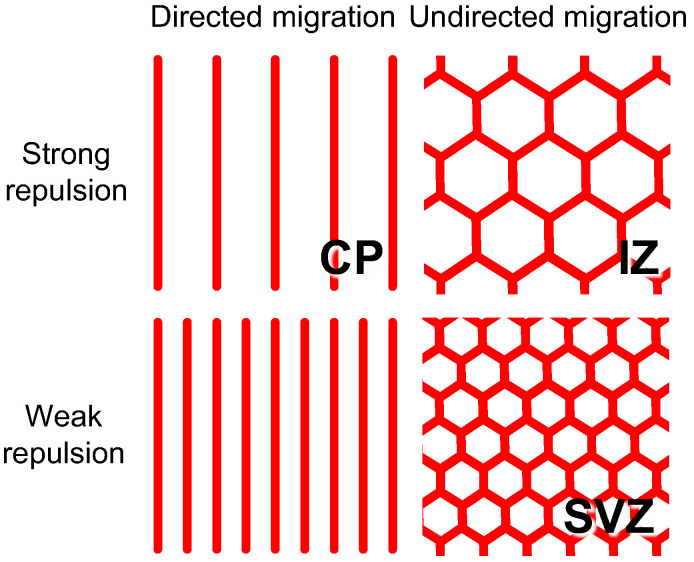
Hypothesis for pattern regulation in periventricular vessels. The characteristic vessel pattens in the CP, IZ, and SVZ were reproduced by a combination of the attractive and repulsive guidance of tip cells in our model. Directed migration controls vessel orientation and represses branch formation. The range and strength of the repulsive interaction determines inter-vessel spacing.

## Data Availability

Not applicable.

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
