# Peer review of "Computational Model Exploring Characteristic Pattern Regulation in Periventricular Vessels"

_life, 2022, doi:10.3390/life12122069_

Round 1
Reviewer 1 Report
There is lack of mathematics, so retitled that paper that will show the main image of your work rather than mathematical model and resubmit the paper.
Reviewer 2 Report
No further comments
Reviewer 3 Report
This study proposed a novel two-dimensional model to describe the angiogenic process, by considering the collective migration of endothelial cell. The model successfully reproduced the different morphologies of bloods vessels in CP and IZ/SVZ. The authors detailedly explain their hypothesis and the biological mechanisms involved in the model, which makes their results convincing. Before publishing, the authors might want to improve their manuscript by considering the following issues.
1. There are many parameters in the computational model. Although I can understand it is difficult to quantitatively determine these parameters in the sense of biology, it might be more convincing if the authors provide some quantitative comparison between experiments and numerical simulations, such as vessel density, or inter-vessel spacing, etc.
2. The interaction between cells and other external forces considered in the model (such as repulsion, external guidance) might induce torques on the ECs, which will result in rotation of the ECs. It seems this has not been considered in the manuscript. Perhaps the authors need to provide suitable justification for this.
3. I suggest authors move the section 3.1 and 3.2 on the numerical model to section 2, i.e., demonstrating the experimental and numerical methods respectively in Section 2.
4. The abbreviation “SP” should be clarified.
5. In fig. 6 and fig.7, it seems in kb and kc, “k” is missing.
6. The use of past and present tenses in the manuscript is somewhat confusing.
7. In equation 5, the repulsion is exerted between head-to-head of ECs. However, In Fig. 2c, the related symbols are not clearly marked, which may cause misunderstanding.
Round 2
Reviewer 3 Report
The authors revised the manuscript carefully. I recommend it for publishing.